# Effect of Grain Size Distribution on Frictional Wear and Corrosion Properties of (FeCoNi)_86_Al_7_Ti_7_ High-Entropy Alloys

**DOI:** 10.3390/e27070747

**Published:** 2025-07-12

**Authors:** Qinhu Sun, Pan Ma, Hong Yang, Kaiqiang Xie, Shiguang Wan, Chunqi Sheng, Zhibo Chen, Hongji Yang, Yandong Jia, Konda Gokuldoss Prashanth

**Affiliations:** 1School of Materials Science and Engineering, Shanghai University of Engineering Science, Shanghai 201600, China; 17805033733@163.com (Q.S.);; 2State Key Laboratory of Advanced Welding and Joining, Harbin Institute of Technology, Harbin 150001, China; 3State Key Laboratory of Materials for Advanced Nuclear Energy, Shanghai University, Shanghai 200444, China; 4Department of Biosciences, Saveetha School of Engineering, Saveetha Institute of Medical and Technical Sciences, Chennai 602117, India; kgprashanth@gmail.com; 5Department of Mechanical and Industrial Engineering, Tallinn University of Technology, Ehitajate tee 5, 19086 Tallinn, Estonia

**Keywords:** spark plasma sintering, heterogeneous structure, grain refinement, frictional wear, corrosion

## Abstract

Optimization of grain size distribution in high-entropy alloys (HEAs) is a promising design strategy to overcome wear and corrosion resistance. In this study, a (FeCoNi)_86_Al_7_Ti_7_ high-entropy alloy with customized isometric and heterogeneous structure, as well as fine-crystal isometric design by SPS, is investigated for microstructure, surface morphology, hardness, frictional wear, and corrosion resistance. The effects of the SPS process on the microstructure and mechanical behavior are elucidated, and the frictional wear and corrosion resistance of the alloys are improved with heterogeneous structural fine-grain strengthening and uniform fine-grain strengthening. The wear mechanisms and corrosion behavior mechanisms of (FeCoNi)_86_Al_7_Ti_7_ HEAs with different phase structure designs are elaborated. This work highlights the potential of using powder metallurgy to efficiently and precisely control and optimize the multi-scale microstructure of high-entropy alloys, thereby improving their frictional wear and corrosion properties in demanding applications.

## 1. Introduction

High-entropy alloys (HEAs) mark a compositional revolution in the design of metallic materials, breaking through the traditional design framework of single-element dominated alloys by combining more than five major elements in a near iso-atomic ratio [1]. This multi-component strategy not only builds an unprecedented compositional design space but also endows HEAs with unique properties, such as high strength, thermal stability, and corrosion resistance [2]. Since Cantor and Yeh proposed the concept in 2004 [3], HEAs have become a research hotspot in materials science, and optimizing their mechanical and chemical properties continues to receive extensive attention.

Spark plasma sintering (SPS) is an advanced powder metallurgy technique [4,5]. The SPS technology integrates plasma activation, hot pressing, and resistance heating, and achieves sintering through the generation of a transient high-temperature field. Compared to commonly used methods for synthesizing HEAs—vacuum melting, mechanical alloying, laser cladding, and selective laser melting [6,7,8,9]—SPS offers distinct advantages. These include rapid heating rates, short sintering times, uniform grain size [10,11], enhanced control over the sintered body’s microstructure [12], and high density [13,14], making SPS a favorable method for HEA production.

The wear and corrosion properties of HEAs have a critical impact on their application as functional materials [15,16,17,18,19,20], and several reports have been published on these topics. This includes microstructural modulation, such as grain boundaries, precipitate–substrate interfaces, and phase distribution in resisting material loss under friction [21]. In particular, precipitation strengthening has emerged as an effective mechanism to increase hardness and prevent wear-induced damage. Ma et al. [22] prepared CoCrFeMnNi/NiCoFeAlTi high-entropy alloy composites by selective laser melting (SLM) and short-time aging treatment, which precipitated orthorhombic CMCM-type (orthorhombic crystalline space group) nano-oxides enriched with Al and Ti and reached an ultimate tensile strength of 731 MPa after aging treatment at 873 K for 2 h. This is attributed to the combined and coordinated effects of precipitation strengthening, dislocation strengthening, and intracrystalline defect-induced high lattice distortion, providing a multi-scale strengthening and hardening mechanism for plastic deformation of high-entropy alloy composites. Since precipitates act as barriers to dislocation motion and grain boundary sliding [23]. Yu et al. [24] fabricated (FeCoNi)_86_Al_7_Ti_7_ HEA by SPS, mainly to investigate the tribological properties of HEA heterostructures by in situ precipitation strengthening. The tribological properties of heterostructured samples were evaluated as a function of the sintering time. The friction coefficient changes from 0.40 to 0.38 when the sintering time is increased from 0 min to 6 min, and its lowest coefficient (0.32) and narrower wear traces are observed at a sintering time of 4 min, indicating better wear resistance. The microstructure of the surface after friction shows the formation of a stable oxide film and columnar nanocrystals, promoting their wear resistance. However, there was no comparison between the heterostructure and equiaxed structure samples. Meanwhile, the stable oxide film formed on the material surface plays a crucial role in their corrosion resistance [25,26]. Yan et al. [27] investigated the corrosion behavior of Al_0.3_CrxFeCoNi HEA with different Cr contents and revealed that the excellent corrosion performance could be attributed to the production of Cr-rich, when the atomic ratio of Cr is relatively high (x = 1.5–2.0), where Cr_2_O_3_ protective surface coating was observed. The effect of heterogeneous structures on the corrosion performance of HEAs is also worth noting.

In this work, (FeCoNi)_86_Al_7_Ti_7_ HEA samples with different powder particle sizes were prepared by SPS. The microstructure, hardness, tribological wear, and corrosion resistance of the HEA were thoroughly investigated.

## 2. Materials and Methods

Aerosolized (FeCoNi)_86_Al_7_Ti_7_ HEA powders (provided by Ningbo Zhongyuan New Material Technology Co., Ltd. Ningbo, China) were used in the present research. To compare the influence of powder morphology, three different powder particle sizes were selected: fine powder with particle size between 0 μm and 30 μm; coarse powder (150 μm to 300 μm); and mixed powder by mixing fine and coarse powders (with a mass ratio of 1:1). The samples after SPS with the above said particle sizes are denoted by S (0–30 μm), L (150–300 μm), and H, respectively. Cylindrical samples with a diameter of 20 mm and a thickness of 12 mm were prepared by SPS (FCT Systeme GmbH HPD-10-SI, Tokyo, Japan) by placing the HEA powders into a graphite mold with graphite foil as the mold inner wall. The SPS was carried out at a temperature of 1303 K (heating rate of 80 K/min) with constant pressure (35 MPa) and a holding time of 8 min. The uniaxial pressure was applied throughout the heating, holding, and cooling processes.

An X-ray diffraction analysis (XRD) system (D/MAX-3BX) with diffraction angle (2θ) scanned in the range of 20°~100° at a scanning speed of 10°/min was used for structural analysis. The surface oxide layer was removed before XRD measurements. All samples were polished with SiC sandpaper up to 2000 girt, followed by further polishing with 1 μm diamond polishing agent (Quzhou Aipu Metrology Instruments Co., Ltd., Quzhou, China) to a mirror finish. For microstructural analysis, samples were etched with a mixture of nitric and hydrochloric acids with a volume ratio of 1:3, and the microstructure of the alloy samples was examined using an optical microscope (OM, Zeiss SteREO Discovery V20, Baden-Württemberg, Germany) and analyzed nearly 1000 grains using ImageJ 1.5.4 software. The elemental compositional distribution was determined using a scanning electron microscope (SEM, EZISS Gemini SEM 300, Baden-Württemberg, Germany) equipped with an energy dispersive spectroscopy (EDS).

An electrochemical workstation (CHI760E, Shanghai, China) equipped with a standard three-electrode system was used to study the corrosion behavior of the (FeCoNi)_86_Al_7_Ti_7_ HEA in 3.5 wt.% NaCl solution at room temperature. A saturated calomel electrode (SCE) was used as the reference electrode (RC), a platinum electrode was used as the counter electrode (CE), and the HEA sample was used as the working electrode (WE). The open-circuit potential (OCP) was continuously monitored as a function of time, and the stabilization of OCP was defined when the potential fluctuation was less than 5 mV within 10 min. Therefore, the experimental procedure began with 60 min OCP test to establish a stable initial state before the potentiodynamic polarization (PDP) test and electrochemical impedance spectroscopy (EIS) test. The initial potential for the kinetic potential polarization and PDP were 1 mV/s and −1 VSCE, respectively. The PDP experiment was terminated at a potential of 1 VSCE. The EIS test was performed on the OCP between the frequency range of 100 kHz to 0.01 Hz with an amplitude voltage of ±10 mV. For the kinetic potential polarization studies, the samples were scanned at 1 mV/s. The fitted circuits demonstrate several components: solution resistance (Rs), double-layer constant phase element (CPEdl), polarization resistance (Rp), inductance (L), and inductive resistance (RL).

The hardness tests were performed using an HXD-1000 digital micro-Vickers hardness tester (Shanghai Optical Instrument Factory, Shanghai, China) with a test load of 20 kgf and a dwell time of 15 s. At least 10 points were measured for each sample, with a spacing of 50 μm between the test points. The functional friction tester (Lanzhou Zhongke Kaihua Technology Development Co., Ltd. Lanzhou, China) with a sliding time of 30 min and a reciprocating speed of 240 rpm was used to evaluate the friction tests in a reciprocating sliding mode with a reciprocating length of 3 mm. YG6 steel balls with a 11.1 mm diameter were used as the mating surfaces according to the ASTM-G133-14 standard [28]. The friction test was performed at room temperature. Wear marks were digitized using a non-contact profiler NANOVEA PS50 3D (Nanovea, Inc. Irvine, CA, USA). The wear losses were measured in mm^3^ using a profiler measurement software, and the wear surfaces were analyzed by SEM-EDS.

## 3. Results

### 3.1. Microstructural Analysis of SPS Sample

The microstructures of the HEA SPS samples (L, S, and H) along with their grain size distribution are shown in Figure 1. The OM images (Figure 1a–c) show that all the samples exhibit typical polygonal grain morphology, with a few microvoidal defects at the grain boundaries. L exhibits a homogeneous coarse-grained microstructure, S exhibits a homogeneous fine-grained microstructure, and H exhibits a heterogeneous structure consisting of specific spatial distributions of ultrafine and coarse grains, i.e., the coarse-grained regions are surrounded by continuously connected ultrafine-grained regions. The number of micropores in H is significantly lower than that in L and S. This can be attributed to the fact that the fine powder effectively fills the gaps of the coarse powder, which promotes the densification of the material, and the shrinkage gradient between the fine powder and the coarse powder decreases during the sintering process, which alleviates the concentration of residual stresses and inhibits the formation of micropores. From Figure 1d–f, it can be seen that the grain size of L ranges from 62 to 315 μm, with an average grain size of 152.57 μm; the grain size of S ranges from 0 to 31 μm, with an average grain size of 11.59 μm; and the grain size of H ranges from 0 to 323 μm, with an average grain size of 72.95 μm. Based on the initial powder grain size of the three samples, the SPS process suppressed excessive coarsening during grain growth, while also preserving the initial microstructural features and microstructural differences of the powders (L and S grain size differences). For the heterogeneous structure design of H, SPS rapid sintering can effectively prevent grain homogenization and make the grain distribution heterogeneous.

Figure 2 shows the XRD diffractograms of the bulk SPS L, S, and H samples. Six diffraction peaks are observed, and the diffraction peaks correspond to face-centered cubic (*FCC*) solid solution ((111), (200), (220), (311), and (222)) and the body-centered cubic (*BCC*) structured L2_1_ phase (ordered intermetallic compound phases) ((110)).

Figure 3a–d shows the low to high magnification SEM images, as well as the EDS energy spectrum of the SPS HEA H sample. In Figure 3a, the overall distribution of grains (from small to large) can be observed. Large voids can be observed at the boundaries of the large grains. As shown by the arrows in Figure 3b, a small number of precipitates are distributed at the grain boundaries, which can be attributed to the L2_1_ phase in combination with the XRD patterns, and part of the precipitated phase was corroded off during the metallographic sampling process to form pores. The sub-structured nano-precipitated phase in the FCC phase is shown in Figure 3c, which is densely interspersed with subgranular grains with an average grain size of 32.67 ± 8 nm. The nanoscale precipitated phase significantly reduces the deformation and wear of the material during friction by impeding dislocation motion, especially at high temperatures and high friction conditions [15]. The EDS maps (Figure 3d) show that the SPS HEA samples a uniform elemental distribution without any aggregation.

### 3.2. Hardness and Wear Performance

The microhardness values observed for the SPS HEA as a function of particle size are shown in Figure 4. The average hardness of the S sample increases by ~6% (421.3 HV) compared with that of the L sample (397.1 HV), whereas the hardness of the heterostructured H samples (397.6 HV) is nearly the same as that of the L samples. The heterogeneous structural design of this alloy sample did not provide a significant gain in hardness.

Figure 5a–f shows the friction and wear data of the SPS HEA samples (L, S, and H) in a dry environment. Figure 5a shows the dry friction coefficient curves of the three types of samples considered. It is observed that the average friction coefficient of the L sample after stabilization is 0.92, which is higher than that of the S sample (0.87). The present results are per the law, where the average friction coefficient is inversely proportional to the hardness. The H sample has a nearly identical friction coefficient curve to that of the L sample at the initial stage of the friction, and the friction coefficient of the H sample decreases afterward. After 15 min, the friction coefficient of the H sample finally reaches a friction coefficient of 0.87. The cross-section profiles of the wear tracks of the three samples are relatively close to each other, and it can be observed that the bottom of the wear track tends to be smoothed gradually with the decrease in the powder particle size. The wear volume losses measured of L, S, H by confocal microscope laser ranging were observed to be 0.734 (±0.015) mm^3^, 0.671 (±0.023) mm^3^, and 0.678 (±0.013) mm^3^, respectively. The wear rates were found to be 8.16 (±0,17) × 10^−3^ mm^3^/(N-m), 7.46 (±0.25) × 10^−3^ mm^3^/(N-m), 7.54 (±0.14) × 10^−3^ mm^3^/(N-m). The specific wear rates of the samples were calculated using the conventional expression (Archard equation):(1)$$Specific\;wear\;rate\;({mm}^{3}\cdot N^{-1}\cdot m^{-1})=\frac{Volume\;loss\;({mm}^{3})}{Normal\;load\;(N)\times Sliding\;distance\;(m)}$$

After the wear tests, the surface morphology and composition of the wear traces as well as the generated debris, were characterized and are shown in Figure 6. It can be observed that the wear surface of sample L shows a concave and convex layering phenomenon (Figure 6a), with a small number of oxides (Figure 6b). A parallel distribution of thin and deep furrows in the form of long strips, and a small amount of plastic deformation in their intervening areas is observed. The main wear mechanisms of L are therefore delamination and abrasive wear, which are consistent with the direction of the sliding (Figure 6c). The wear surface undergoes plastic deformation and fracture under cyclic loading, gradually forming a relatively smooth surface. Subsequent relative sliding of the friction pair, the surface layer undergoes shear plastic deformation and accumulates dislocations [29], resulting in the increased dislocation density close to the friction surface. The wear mechanism of sample S is roughly similar to that of sample L, but due to the smaller particle size, the denser dislocations induce a fish-scale plastic deformation of the worn surface (Figure 6f), which results in the removal of the abrasive debris in the form of a smaller, powdery chip.

The wear surface of H sample is mainly composed of powdery abrasive debris mixed with a few massive abrasive debris (Figure 6h), which is caused by the exfoliation of different sized grains in the heterogeneous structure under cyclic loading. The fine powdery abrasive debris oxidizes under cyclic loading and adheres to the grooves in the wear surface layered with abrasive debris under loading, thus forming a new surface oxide layer (Figure 6i). The enrichment of elemental oxygen in the EDS diagram corroborates the presence of an oxide layer (Figure 7c, Table 1). Multi-element oxides (e.g., oxides containing aluminum, titanium, etc.) usually exhibit low coefficients of friction, which enable them to form lubricating layers on the friction surfaces to reduce the direct contact between the metal substrate and the mating surfaces, which in turn slows down the wear of the samples by cyclic loading, which reveals the reason for the small decline in the coefficient of friction of the H samples after 15 min and the lower wear rate relative to the L samples (Figure 5a,c). At the same time, this oxide layer will again produce exfoliation and be adhered by new powder abrasive debris during relative sliding with the friction partner, so the H sample is moving from abrasive to oxidized wear and finally to a wear mechanism with a mixture of adhesive and oxidized wear. Therefore, it can be inferred that the design of the heterogeneous structure moderates the transition process of the wear mechanism to a certain extent and enhances the wear resistance of the material.

### 3.3. Electrochemical Corrosion Properties

Figure 8a,b shows the OCP and kinetic potential polarization curves (recorded at a scan rate of 1 mV/s) of SPS (FeCoNi)_86_Al_7_Ti_7_ HEA in 3.5 wt.% NaCl solution. It can be seen from Figure 8a that the open-circuit potentials of the three samples all tend to stabilize with the increase in the immersion time. The E_ocp_ of L, H, and S HEAs after immersion for 3600 s are −325 mVsce, −388 mVsce, and −383 mVsce, respectively. It can be observed that the H samples show similar stabilizing potentials to those of the S samples, implying that the passivation films have better stability and protection. This is attributed to the increase in grain boundary area, which in electrochemistry is more likely to adsorb ions and reactive substances in solution, giving the material surface a higher cathodic potential. Generally, anodic dissolution is an oxidizing process in which the metal loses electrons, whereas a higher cathodic potential hinders the flow of electrons from the metal substrate to the solution, which inhibits the anodic reaction from proceeding and reduces the corrosion rate of the metal [30].

Figure 8b shows the polarization curves obtained under OCP conditions, and the results of the kinetic potential polarization tests are shown in Table 2. The corrosion rate, E_corr_ (corrosion potential), and I_corr_ (corrosion current density) were calculated from Tafel curves. Figure 8b shows that the corrosion behavior of the H and S samples are similar, and the values of E_corr_ for the two samples (H and S) are −0.6404 V and −0.6333 V, respectively, both of which are higher than the E_corr_ of −0.7492 V observed for the L sample. However, the highest corrosion current density of 11.62 μA/cm^2^ was observed for the sample S (among the three considered conditions) and is attributed to the fact that sample S has the largest number of grain boundaries. The grain boundaries act as preferential dissolution sites and are more susceptible to redox reactions under the action of an applied electric field. Both grain size and phase composition have been reported to affect the corrosion behavior of HEAs [31]. The two samples (H and S) had similar phase compositions, while the heterostructured H sample achieved a high corrosion resistance similar to that of the grain-fine S sample, indicating that the design of the heterostructure achieves the purpose of stable formation of the passivation film by reducing the number of grain boundaries. This is consistent with the result of the formation of fine grains.

Figure 9 shows the EIS results of the EIS characterizing the passivation film on the surface of the alloy, the electrode surface properties, and the corrosion resistance of the SPS (FeCoNi)_86_Al_7_Ti_7_ HEA. In Figure 9a, all Nyquist plots show that the HEAs are characterized by capacitive loops in the high- and medium-frequency ranges and inductive loops in the low-frequency range. The capacitive loops are related to the charge transfer process. It was found that the radius of the capacitive arc is enlarged for H sample with a heterogeneous structure compared to L sample, which represents a higher corrosion resistance, by comparing the capacitive reactance arc diameters of the three curves (related to the polarization resistance Rp). Equivalent circuit diagrams were fitted to the EIS data using ZView2 software as shown in Figure 9b, and the fitted calculated data are shown in Table 3. The inductance loop observed in the low-frequency region corresponds to the inductance (*L*) and its associated inductive resistance (*R_L_*) in the equivalent circuit model shown in Figure 9b. The presence of inductance (*L*) suggests that the material surface is in a non-ideal passivated state. Furthermore, the higher *R_L_* value obtained for sample H (Table 3) indicates greater stability of its passive film, requiring more energy to initiate its breakdown [32]. Since the surface of the HEAs sample is not completely flat, CPEdl is used instead of capacitor (C). *n* is a parameter that describes the surface roughness; when *n* is 1, the surface is completely smooth and flat, and when a value of *n* is 0, it indicates a rough surface. According to the EIS fitting results in Table 3, the heterogeneous structure and grain refinement led to an increase in the impedance values of *R_p_* and *R_L_*, which indicates an increase in the difficulty of transferring charge particles, and both methods led to a decrease in the value of CPEdl. Since the thickness of the passivation film is inversely proportional to CPEdl, so the design of the heterogeneous structure leads to the formation of the passive film, which improves the corrosion resistance. Figure 9c,d shows that H and L have larger phase angle values and larger impedance moduli over a wide frequency range, thus corroborating that heterogeneous structures and grain refinement can promote the formation of more stable and corrosion-resistant passivation coatings in samples.

## 4. Summary

This study investigates the microstructure evolution, microhardness, friction and wear properties, and corrosion resistance of (FeCoNi)_86_Al_7_Ti_7_ HEAs prepared by SPS in three different particle size powder configurations. The main findings are as follows:(1)(FeCoNi)_86_Al_7_Ti_7_ HEAs are mainly composed of a face-centered cubic (*FCC*) solid solution phase and a body-centered cubic (*BCC*) structure L2_1_ phase. The surface morphology of L and S samples show equiaxial grains, and the average grain size decreases from 152.57 μm to 11.59 μm due to the difference in the initial powder grain size, while the microstructure of H exhibits a heterogeneous structure consisting of specific spatial distributions of ultrafine and coarse grains, i.e., the coarse-grained regions are surrounded by continuously connected ultra-fine-grained regions.(2)Due to the fine-grain reinforcement, S exhibits the highest hardness (421.3 HV_0.2_), the lowest average coefficient of friction (0.87) and the lowest wear rate (7.46 ± 0.14 × 10^−3^ mm^3^/(N·m)), and the wear mechanisms operating in both L and S are delamination and abrasive wear. The H sample with heterogeneous structure, on the other hand, achieves similar wear resistance to the S sample through a mixed wear mechanism of adhesive and oxidative wear.(3)The results of electrochemical corrosion experiments show that all three HEA samples exhibit obvious phenomenon of passivation, and H exhibits strong corrosion resistance similar to that of S. The corrosion potential of H increases to −0.6404 V, the corrosion current density is 8.39 μA/cm^2^.

## Figures and Tables

**Figure 1 entropy-27-00747-f001:**
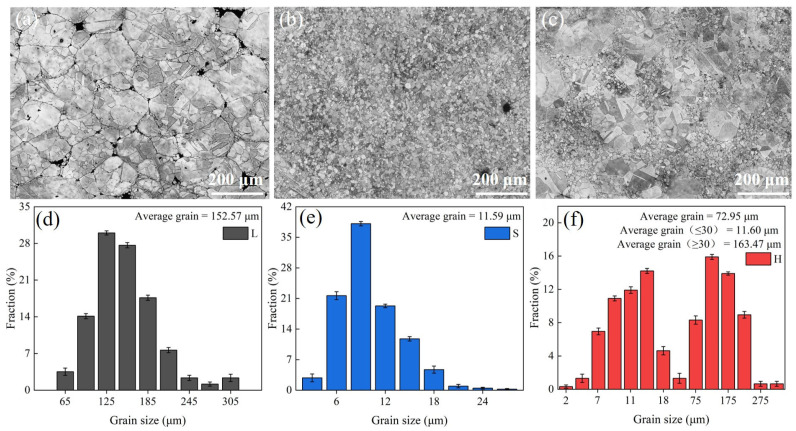
Optical microscopy images showing the grain morphology in the three considered high-entropy alloy (HEA) samples after spark plasma sintering (SPS): (**a**) L, (**b**) S, and (**c**) H, respectively. The grain size distribution plots for the SPS HEA samples: (**d**) L, (**e**) S and (**f**) H.

**Figure 2 entropy-27-00747-f002:**
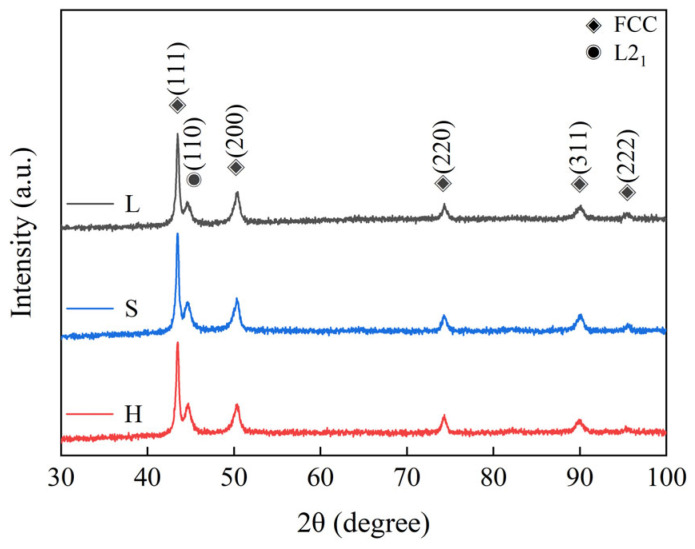
X-ray diffraction pattern of the spark-plasma-sintered (FeCoNi)_86_Al_7_Ti_7_ high-entropy alloy as a function of particle size.

**Figure 3 entropy-27-00747-f003:**
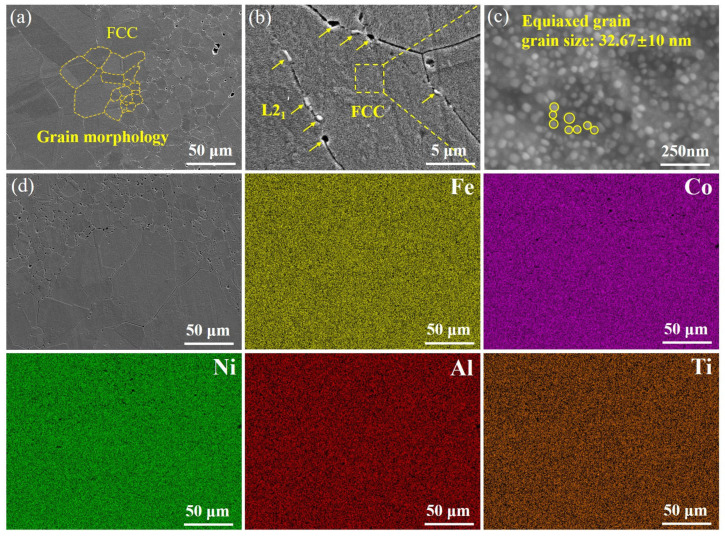
(**a**–**c**) The low to high magnification SEM images; (**d**) SEM map with EDS elemental mapping distribution (Fe, Co, Ni, Al, and Ti) of the corresponding region of H samples.

**Figure 4 entropy-27-00747-f004:**
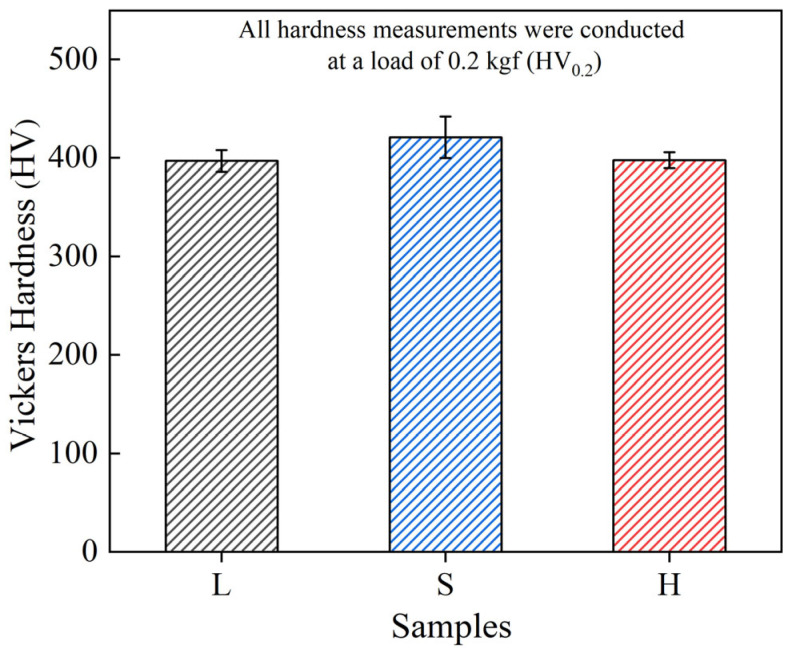
The hardness of spark-plasma-sintered (FeCoNi)_86_Al_7_Ti_7_ high-entropy alloy as a function of particle size.

**Figure 5 entropy-27-00747-f005:**
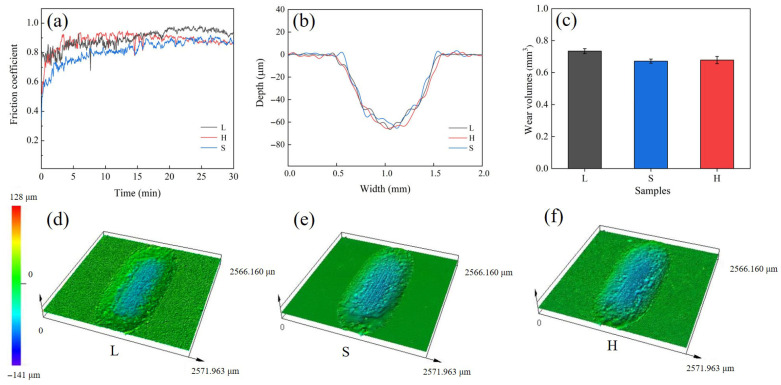
(**a**) The friction coefficient as a function of time for three (FeCoNi)_86_Al_7_Ti_7_ high-entropy alloy samples. (**b**) Two-dimensional cross-sectional depth profiles of wear marks. (**c**) The wear volume as a function of particle size for three (FeCoNi)_86_Al_7_Ti_7_ high-entropy alloy samples. (**d**–**f**) Three-dimensional surface profiles of the wear traces.

**Figure 6 entropy-27-00747-f006:**
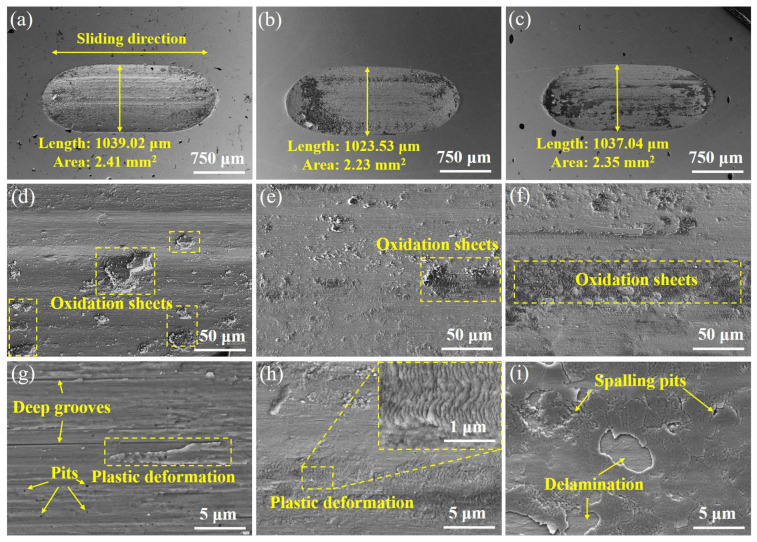
(**a**) The scanning electron microscopy (SEM) images of the wear trace for the sample L. (**b**) and (**c**) Magnified SEM images showing the wear traces for the sample L. (**d**) The SEM image showing the wear traces for the sample S. (**e**,**f**) Magnified SEM images showing the wear traces of the sample S. (**g**) The SEM image showing the wear trace for sample H. (**h**,**i**) Magnified SEM images showing the wear traces for the sample H.

**Figure 7 entropy-27-00747-f007:**
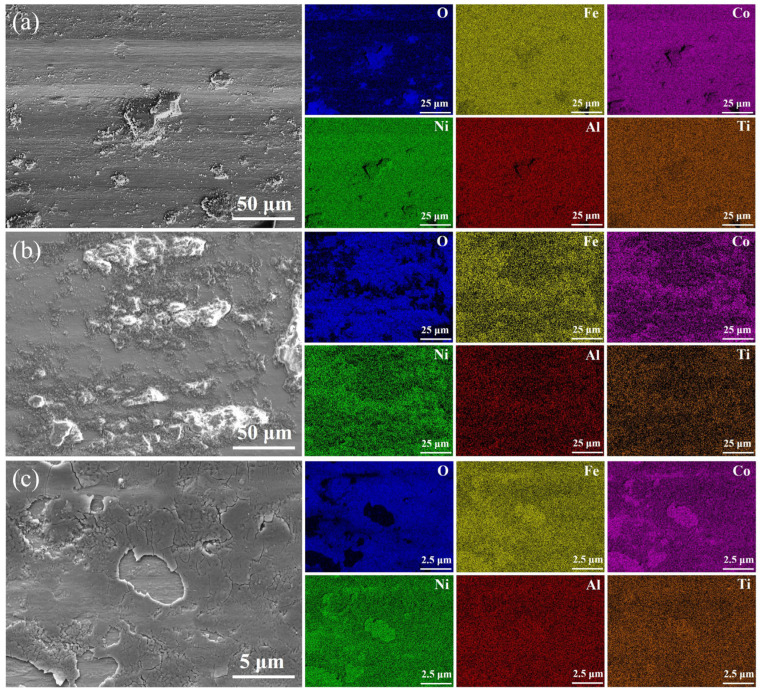
The scanning electron microscopy images and the energy dispersive maps (O, Fe, Co, Ni, Al, and Ti) along the wear surface for the spark-plasma-sintered (FeCoNi)_86_Al_7_Ti_7_ high-entropy alloys as a function of particle size: (**a**) L, (**b**) S, and (**c**) H.

**Figure 8 entropy-27-00747-f008:**
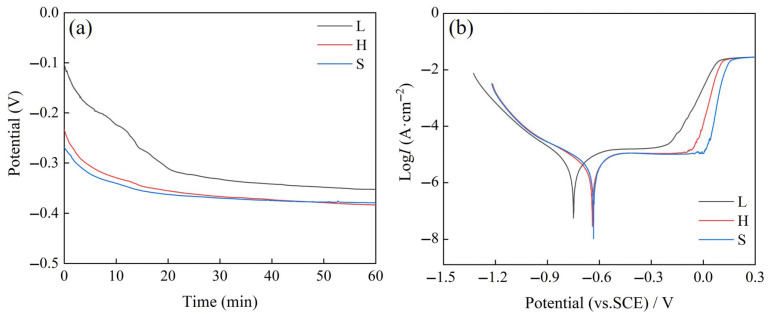
(**a**) Open-circuit potential as a function of time (min), (**b**) Kinetic potential polarization of the spark-plasma-sintered (FeCoNi)_86_Al_7_Ti_7_ HEA.

**Figure 9 entropy-27-00747-f009:**
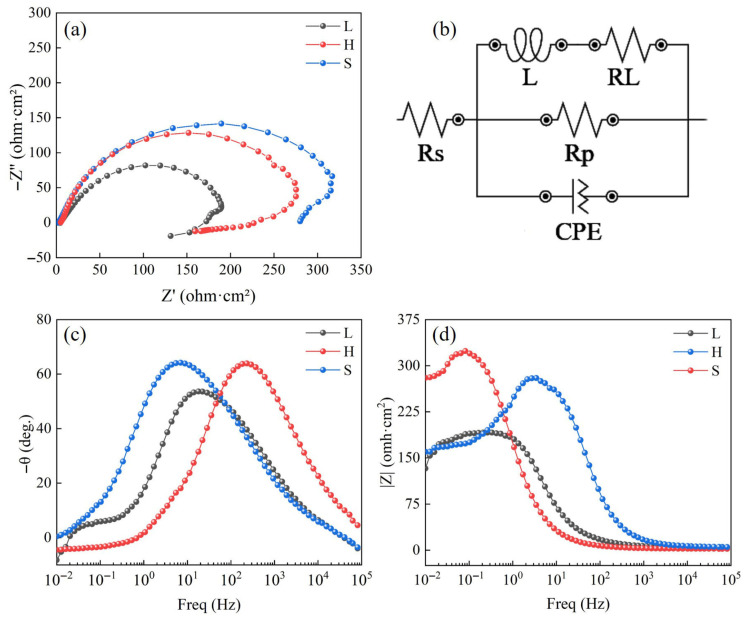
Electrochemical impedance behavior of the spark-plasma-sintered (FeCoNi)_86_Al_7_Ti_7_ HEA at the OCP stable state and corresponding fitting results of EIS data in 3.5 wt% NaCl solution. (**a**) Nyquist plots, (**b**) Equivalent circuit plots and (**c**,**d**) Bode plots.

**Table 1 entropy-27-00747-t001:** EDS analysis results for different samples in Figure 7.

Sample	Content of Elements (at%)					
	O	Fe	Co	Ni	Al	Ti
L	12.17	25.42	25.43	24.65	6.13	6.21
S	46.29	15.47	15.25	14.98	4.20	3.81
H	45.80	15.56	15.23	14.68	3.85	3.87

**Table 2 entropy-27-00747-t002:** The electrochemical parameters fitted from the potentiodynamic polarization curves of the spark-plasma-sintered (FeCoNi)_86_Al_7_Ti_7_ HEA measured in 3.5 wt.% NaCl solution.

Sample	E_corr_ (V)	Icorr (μA/cm^2^)
L	−0.7492	8.32
S	−0.6333	11.62
H	−0.6404	8.39

**Table 3 entropy-27-00747-t003:** The equivalent circuit parameters for the (FeCoNi)_86_Al_7_Ti_7_ spark-plasma-sintered samples calculated by fitting the impedance spectra.

Sample	Rs (Ω·cm^2^)	CPEdl (10^−6^ F·cm^−2^)	*n*	Rp (10^3^ Ω·cm^2^)	L (10^3^ Hz·cm^2^)	RL (10^3^ Ω·cm^2^)
L	4.7	0.9	0.87	222.6	342.4	289.4
S	2.8	0.7	0.83	373.6	3243	664
H	5.6	0.7	0.82	312	59.8	350.7

## Data Availability

Data is contained within the article.

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
