# Peer review of "Effect of Grain Size Distribution on Frictional Wear and Corrosion Properties of (FeCoNi)86Al7Ti7 High-Entropy Alloys"

_entropy, 2025, doi:10.3390/e27070747_

Round 1

Reviewer 1 Report

Comments and Suggestions for Authors

This study investigates the microstructure, mechanical properties, and corrosion behavior of (FeCoNi)₈₆Al₇Ti₇ HEAs processed via SPS using three powder size distributions. The work addresses a relevant topic in HEA design, particularly the role of heterogeneous structures in wear/corrosion resistance. While the experimental methodology is generally robust and the data support some conclusions, the manuscript suffers from significant inconsistencies in composition notation, insufficient methodological details, superficial mechanistic analysis, and poor presentation of results. Major revisions are required to address these flaws before consideration for publication.

---The abstract and introduction cite "(FeCoNi)₈₆Al₇Ti₇," yet Sections 2–3 repeatedly use "(FeCoNi)₅₆Al₇Ti₇" (e.g., p. 2) and "(FeCoNi)₉₈Al/Ti₂" (p. 3, 6). This fundamental error casts doubt on the entire study’s validity. Authors must verify and standardize the alloy composition throughout.

---Table 1 reports corrosion rates in "numpy" (p. 9)—a nonsensical unit undefined in the field. Corrosion rates are typically in mm/year (mm/y) or mils per year (mpy). Correct units and clarify calculation methods.

---The corrosion tests lack critical parameters: No surface preparation protocol (e.g., polishing grit, final roughness). OCP stabilization criteria unstated (p. 8). PDP scan rates omitted (p. 3 claims 1 mV/s, but p. 9 states "kinetic potential polarization" without specifics). Provide full experimental protocols to ensure reproducibility.

---Wear volumes (p. 6) are reported without describing the formula used (e.g., ASTM G133). The claim that wear rate is "inversely proportional to hardness" conflicts with the H-sample data (hardness ≈ L-sample, yet wear rate closer to S-sample). Justify calculations and discuss anomalies.

---The "H" sample’s heterogeneous structure shows no hardness improvement vs. "L" (Fig. 4), yet enhanced wear/corrosion resistance is attributed vaguely to "mixed wear mechanisms" and "oxide layers." No TEM/HR-STEM evidence supports nanoscale precipitate effects (p. 5). Provide crystallographic data (SAED, EBSD) to substantiate strengthening mechanisms.

Other comments:
Figure 3c: Claims "sub-granular grains" (avg. 32.67 nm) but shows no scale bar. Add scale bars to all SEM/TEM images.

Figure 4: Label axes clearly (hardness units: HV₀.₂?).

Section 3.1: XRD (Fig. 2) shows FCC+BCC peaks, but phase fractions are unquantified. Include Rietveld refinement data.

p. 6: "Friction coefficient of H-sample finally reaches -0.87" – negative friction? Correct typo.

p. 7: Wear debris analysis (Fig. 6) lacks quantitative EDS (e.g., O content in oxides).

Table 2: Units for CPEdl (μF·cm⁻²) and RL (kΩ·cm²) are inconsistent. Standardize.

p. 9: EIS fitting uses an inductive circuit (L, RL) but fails to justify its relevance to HEA passivation. Clarify.

References: Multiple formatting errors (e.g., Ref. 4–9 brackets "[4][5]" vs. "[4–9]"; Ref. 23 incomplete journal title).

Abbreviations: Define "CMCM-type" (p. 2) and "L2₁ phase" upon first use.

Discussion: Weak linkage to prior art (e.g., Yu et al. [24]’s similar work). Contrast findings explicitly.

Reviewer 2 Report

Comments and Suggestions for Authors

Dear Authors, 

Thank you for your valuable work. I think, your manuscript is very well written, the reading is easy ant the results are presented in a correct way. I have some general questions and some suggestions. 

  1. There is a typo in line 135: boundaries.L 
  2. Figure 2: I see only one identified peak for L21 phase. This is strange for me. Based on your calculations, how many peaks should be visible on this XRD? 
  3. Figure 3: when finishing the polishing process, one can find residual polishing agent on the surface. The figure 3 c resembles to silica nanoparticles. I see that you used diamonds, but can you please provide the exact type of your polishing agent? 
    If the case is, that these small particles ara grains, you should find a thermodynamic reason to conclude them.
  4. Typo in line 253: Figure 8a,b
  5. Figure 9: please use mm/year instead of mmpy

As you can see, question 2 and 3 are serious, the others are only comments. I am waiting for your response. 

Round 2

Reviewer 1 Report

Comments and Suggestions for Authors

No more comments.

Reviewer 2 Report

Comments and Suggestions for Authors

Thank you for the improvements.